# Effect of guided dual-sensory information on motor learning outcomes based on spatiotemporal dimensions

Liwa Sha[1,2], Wen Hsin Chiu [3]*

1 Department of Sports Training, Jilin Sport University, Changchun, Jilin, China, 2 Department of Education and Learning Technology, National Tsing Hua University, Hsinchu, Taiwan, 3 Department of Kinesiology, National Tsing Hua University, Hsinchu, Taiwan

* whchiu@mx.nthu.edu.tw

## Abstract

The effectiveness of instructional information is crucial to enhancing motor learning outcomes. However, few studies have explored the mechanisms of augmented attention-guiding dual-sensory information. Based on the spatiotemporal dimensions of movement staging and movement limb segments, the current study analyzed how guided dual-sensory information affects learning a basketball set shot. A teaching experiment involving 132 middle school students was used to analyze three instructional methods: visual-only (video), visual–auditory (video with narration), and guided dual-sensory information with visual cues (video with narration and markers). Participants learned set shooting over three weeks. The results revealed that (1) All three groups (visual-only, dual-sensory, and dual-sensory with visual markers) exhibited improved movement result performance after training, with the group receiving guided dual-sensory instruction exhibiting substantially superior posttest movement result performance than the group receiving video alone. (2) audiovisual (dual-sensory) information considerably enhanced movement pattern performance, and dual-sensory information with additional visual markers strengthened movement pattern retention and optimized upper-limb movement pattern performance during the preparation stage. (3) Both the group receiving video with narration and the group receiving guided dual-sensory instruction exhibited substantial improvements in motor cognition, with the dual-sensory information with visual cues associated with the strongest facilitation effect on upper-limb cognition during the execution stage. These results reveal that guided dual-sensory information substantially enhances motor learning outcomes in basketball shooting, with required movement guidance of the upper-limb segments being greater than that for other body segments. The findings indicate that for digital instruction in physical education, instruction should include strategically employed guided attention methods and multidimensional combinations of information presentation. This study provides a novel perspective on motor learning and a spatiotemporal framework for understanding the learning process.

**Data availability statement:** All relevant data for the research results described in the manuscript have been stored on the public website figshare (Doi: https://doi.org/10.6084/m9.figshare.30023368). Activate DOI after publication of the paper. Currently providing a private link for editorial review. https://figshare.com/s/f12c6d902081ccc362c9.

**Funding:** Supported by the Independent Innovation Fund of Jilin Sport University.

**Competing interests:** The authors have declared that no competing interests exist.

## Introduction

In recent years, motor learning in physical education has been trending toward digitalization. Studies have demonstrated that using multimedia to target different senses can improve learning outcomes, gradually uncovering the importance of guided attention in instruction [1–3]. Nevertheless, whether digital tools can lead to lasting changes in motor skills remains unknown [4], because the current research status neglects the need to analyzing action essentials from the spatiotemporal dimension of motion. This neglect poses difficulties for physical education teachers using digital teaching applications because they cannot determine which method of presenting information is most effective for staging individual limb segment movements. The present study used Mayer's cognitive theory of multimedia learning (CTML) as its framework. This theory posits that when learners receive information through properly designed visual and auditory channels, this information reduces extraneous cognitive load, promotes deep processing, and enhances motor learning outcomes [5]. On the basis of the dual-channel signaling principle of the CTML [6], this study proposed a visual＋audio＋markers guided information presentation strategy. By marking key movement nodes, this strategy reduces extraneous cognitive load on the learner of a basketball set shot. Furthermore, this study examined changes in motor learning outcomes by analyzing performance across the spatiotemporal dimensions of movement staging and limb segments. Addressing these questions is crucial for improving teaching materials and supporting digital teaching in physical education.

Information is key in motor learning because perception and movement influence each other. Most sensory information for skilled actions is obtained from the environment, primarily through vision and hearing. Vision enables learners to engage in observational learning, and hearing enables them to receive verbal instructions [7–9]. Therefore, acquiring external information is essential for motor learning. Demonstrations, videos, verbal cues, and other prompts are helpful for learning [10]. In the early stages of motor learning, the human mirror neuron system plays a crucial role in enabling cognitive imitation. Who demonstrates the motor skills being taught, the method of demonstration, and how additional information about these skills is provided are essential considerations in the provision of instruction [11]. Multimedia instruction supported by technology can enhance learning by providing a continuous flow of information learner centered [5,12]. Employing such instruction can overcome the limitations of traditional physical education, where information delivery is often restricted by biological, environmental, and task-related factors [13]. For example, one study reported that video observation considerably enhanced elementary school students' long jump skill motor learning outcomes [2]. Video observation employs external information to guide learners to focus on their movements, activating the mirror neuron system to achieve automated "observation to imitation"and reduce the cognitive load required to maintain internal control. Such external visual cues efficiently promote the acquisition and transfer of motor skills by enhancing perception–action coupling. However, media are merely the means of delivering content; the quality of the content is what truly affects learning outcomes [14]. Therefore, teaching materials must effectively convey information.

Cognitive load theory focuses on how an individual allocates mental resources and suggests that insufficiently accounting for memory limits in instructional design can hinder learning [15,16]. Mayer combined cognitive load theory with dual code theory, creating the CTML [17]. Dual code theory suggests that individuals process information through both images and language; that is, learners receive instructional information through different sensory channels [18]. Multimedia instructional design should reduce unnecessary cognitive load and promote understanding of material [5]. Key elements of multimedia learning materials include text, video, audio, images, and animation, with their use varying in accordance with learning needs [17,19]. The most effective method for delivering multimedia content differs with the subject and learning context.

In the field of physical education, dynamic information is widely considered more effective for achieving motor learning than static information is. For example, a study on judo skill learning revealed that dynamic representations produced better learning outcomes than did static images [20]. Another study demonstrated that dynamic visual aids improved learning outcomes for judo referee gestures and encouraged the development of positive learning attitudes [21]. Research has indicated that dynamic video instruction can quickly improve motor performance in hurdling skills [22], breaststroke skills [23], badminton skills [24], and significant improvements in the kinematic parameters of weightlifting snatch technique [25]. Additionally, adding narration to videos for learning basketball skills was reported to positively affect learning outcomes [26], highlighting the benefits of high-quality commentary [27]. Learners receive information through different sensory channels [18], and dynamic audiovisual content is helpful for learning when learning material is based on human movement [28].

Research suggests that employing multimedia methods for multisensory learning can boost motivation [5] and improve cognitive learning outcomes. Nevertheless, whether such methods improve skill performance remains unclear [20]. In research regarding gymnastics, video instruction was linked to better motor learning, motivation, and self-assessment in early childhood physical education [1]. In addition, multimedia use in teaching shot put improved self-efficacy and perceived learning outcomes but did not enhance motor learning outcomes or enjoyment [29]. Vernadakis et al. reported that traditional teaching methods and multimedia instruction yielded similar learning outcomes in teaching adolescent basketball shooting techniques [30]. Another study demonstrated that a video intervention exerted a positive effect exclusively on training motivation in adolescent basketball players [31]. Motor learning requires simultaneously processing multichannel information and executing physical movements; these processes compete for limited resources in the learner's working memory [15,32]. When the information in action learning materials lacks proper design and organization, both information capacity processing and motor performance are limited, and redundancy effects will be amplified [32]. The aforementioned findings highlight the need for carefully designed and organized motor learning materials to ensure effective information processing and optimal performance.The effectiveness of observational learning in motor skills is determined by a learner's ability to focus on key aspects of a movement. Causer et al. indicated that using instructional markers helped novice basketball players quickly grasp key information, which improved their shooting skills [33]. Additionally, research revealed that coach-directed gaze guidance helped beginners focus on relevant elements of motor skills [3]. These findings highlight a key challenge of multimedia instruction: in multimedia presentations, experienced athletes often exhibit more efficient gaze patterns when faced with complex movements, frequently observing task-relevant areas rather than extraneous ones [34]. Novices tend to miss crucial information, which influences the effectiveness of their learning [33]. Guiding learners attention to effective information while reducing extraneous cognitive load caused by searching for information during learning, requires the provision of appropriately focused knowledge on movement to essential information and cognitive load should be reduced. Cognitive load should match the cognitive demands of motor learning. Reducing cognitive load during motor learning requires providing information to guide learner attention and facilitate the construction of mental models [15]. Overloading working memory with too much or overly complex information can prevent learners from achieving optimal performance [34]. According to resource allocation theory, mental operations are limited by the availability of resources and data. Therefore, instructional materials should direct attention, manage limited attentional capacity, and reduce task competition to enhance learning and performance [15,16].

The current study's review of the literature revealed two main findings. First, a consensus has yet to be reached regarding the method of presenting information that can most effectively improve motor learning, with the composition of the methods that have been employed varying considerably. Research has not yet systematically examined methods involving sensory channels expansion and progressive implementation of attentional guidance. Cognitive load theory and the dual-channel with signaling principle of the CTML suggest that motor learning requires matching sensory channel capacity with attention-guiding design [6,15]. Second, the methods that have been employed for assessing motor learning outcomes have differed. Few studies have analyzed cognitive and performance outcomes on the basis of movement stages and limb segments. Because movement is continuous and movement perception relies on sensory cues, breaking down actions is essential to understanding and remembering them [35–37]. Movement staging refers to dividing motor skills into consecutive logical phases on the basis of their temporal sequence and technical structure to facilitate systematic learning, training, and correction [38]. Movement staging is a fundamental instructional, training, and analytical tool in motor learning. Fitts' three-stage model of movement staging [39] posits that decomposing movements to reduce cognitive load is critical to skill acquisition. Additionally, Schmidt's schema theory suggests that phased practice establishes refined motor schemata and enhances movement adaptability [40]. Studies have verified that step-by-step practice reduces cognitive load in beginners, optimizes learning efficiency, and accelerates error correction through phased feedback [41,42]. However, movement staging is not a fixed, inflexible process; instructors may divide phases according to instructional needs, constructing cognitive frameworks through phase decomposition to enhance learning efficiency.

Studies that have focused on movement stages or kinematic parameters in multimedia teaching are rare. Kyriakidis et al. reported that video instruction in long jump improved performance in areas such as leg joint angle and trunk tilt, with such instruction aiding in learning takeoff techniques [2]. Additionally, Pastel et al. discovered that virtual reality and video methods were equally effective for teaching, in karate movements after four sessions, for example key factors such as study time (pre-, post-, retention) and limb segments (upper body, lower body, fist posture) [43]. Movement patterns are sequences of actions with specific spatial and temporal arrangements [4]. Segmenting motor learning events can help with understanding, predicting, and learning actions. The above research indicates that suggests that the relationship between information presentation and motor learning might be obscured by a lack of detailed analysis of limb segments and movement staging and overly broad evaluation metrics. This may also be the reason why the relationship between learning outcomes in different sports programs is still unclear, namely the lack of exploration into the spatiotemporal dimensions of motor performance and motor cognition.

The set shot is the main scoring method in basketball and is central to offensive and defensive strategies [44]. Improving the teaching and practice of shooting techniques is essential for learning basketball. The set shot is a highly visual motor skill [44] that requires learners to establish internal representations by observing the movements, a process consistent with the fundamental assumptions of observational learning and imitation in multimedia learning theory. Additionally, this skill exhibits a well-defined spatiotemporal movement structure that can be clearly divided into preparation stage and execution stages and principally involves control of the upper-limb, the lower-limb, and the ball-handling. These characteristics render the set shot particularly well suited to examining the effects of movement staging and limb segment information guidance in phased motor skill instruction [30,44]. Finally, the set shot is a fundamental skill in secondary school physical education. Students typically lack experience with this skill, and its stable movement characteristics facilitate standardized assessment and instructional application. Current materials for learning the set shot do not involve diverse sensory channels or provide cues and attentional guidance for beginners. Current materials also fail to consider variations in movement stages and specific limb segments.

The recognition of these deficiencies in set shot pedagogy prompted the following research question: Which of the three information presentation methods—visual-only (video), dual-sensory audiovisual (video with narration), and guided dual-sensory instruction (video with narration and markers)—is most effective in motor learning? To address this question and optimize motor learning outcomes, this study employed an instructional experiment grounded in the CTML [5].

Basketball shooting instructional information was provided to three groups of seventh-grade students through three information presentation methods: video only, video with narration, and video with narration with markers, to investigate how different information presentation methods influence shooting skill acquisition. This study hypothesized that incorporating verbal narration and visual markers (red flashing frames) in instructional videos would enhance learners' focus on critical motor learning information, increasing movement pattern performance and motor cognition outcomes. This design is consistent with the core proposition of CLT, which posits that learners most effectively manage limited cognitive resources by focusing on the most task-relevant information. Furthermore, on the basis of theories of motor learning, this study conducted analyses of spatiotemporal data derived from the movements and spatial dimensions of limb segments during set shot exercises to explore how information presentation affects learners' performance and motor cognition.

The insights gained from this study can help with understanding students' behavior during shooting practice, improve their performance, and render learning more enjoyable. By identifying the optimal combination of presentation methods for shooting instruction, this study contributes to enhancing digital instructional materials, addresses the lack of research into digital motor learning, and improves the effectiveness of physical education.

## Methods

### Participants

This study recruited 132 seventh-grade students from a junior high school in Changchun, China. The average age of the participants was $13 \pm 0.6$ years. The study school admits students using a proximity-based enrollment policy and maintains regular mixed-gender classes. Under these conditions, the motor abilities of the students in the same grade approximate a normal distribution, satisfying random sampling criteria. We employed cluster sampling, randomly selecting three out of the nine seventh-grade classes. The participants were randomly assigned to one of three instructional conditions: a video group ($n = 44$), a video with narration (audiovisual) group ($n = 44$), and a video with narration and markers (guided dual-sensory instruction) group ($n = 44$), with Classes A (video) and B (video with narration) each contributing 22 male and 22 female students, and Class C (video with narration and markers) contributing 21 male and 23 female students. During the experimental design phase, consultation with physical education teachers and a review of historical class records verified that the selected classes exhibited no significant differences in physical literacy, had comparable average physical education test scores, and exhibited equivalent fundamental motor abilities. The participants had no prior basketball learning experience, and the school's physical education curriculum had not yet covered set shot skills. None of the participants had any conditions that could have affected their motor skill learning or cognitive abilities. Before the experiment, all participants and their parents/guardians were fully informed of the study procedures and safety considerations, and written informed consent was obtained from both the participants and their parents/guardians. Three experts independently conducted baseline assessments of the participants' motor skills. As detailed in the Results section, no significant between-group differences were observed in the pretest set shot performance among the three groups ($p > 0.05$), ensuring sample homogeneity. Each group was taught using one of three multimedia methods during physical education classes: presentation of visual-only information, visual and auditory information, and guided dual-sensory information with markers. The content of the lessons focused on learning the set shot. During the experimental period, the participants did not engage in any additional basketball activities or any form of basketball skills training outside the study protocol. This study was approved by the Scientific Research Ethics Committee of Jilin Sport University (No. 20240820−1).

On the basis of the recommendations of Peng et al. regarding sports science experimental samples [45], G*Power 3.1.9.7 was employed to calculate statistical power using the following parameters: significance level ($\alpha = 0.05$), total sample size ($N = 132$), mixed-design analysis of variance (ANOVA) of factors of group (video vs. video with narration vs. video with narration and markers) × time (pretest vs. posttest vs. retention test), detecting a medium effect size ($f = 0.25$). The achieved power of 0.87 met the 0.8 threshold [46], demonstrating that the experimental design had adequate sensitivity to detect significant interaction effects between instructional methods and timepoints.

## Design of experiment

The instructional experiment of this study involved three types of multimedia methods, which served as independent variables: presentation of a video, a video with narration, and a video with narration and markers. Instruction was provided using one of three modalities: visual-only information, visual and auditory information, and dual-sensory information with markers. These methods were examined across movement stages and limb segments to assess their effects on learning the set shot. The dependent variables were motor cognitive related to understanding the set shot and motor performance related to movement pattern performance and movement result performance.

Movement patterns and movement performance were assessed using a mixed design ANOVA, with between-group factors (video, video with narration, and video with narration and markers) × within-group factors (pretest, posttest, retention test). Motor cognition was evaluated using a posttest-only design.

## Research instruments

**Set shot motor learning materials.** This study used instructional videos that featured the set shot. These videos were edited in Final Cut Pro. All three groups watched videos of the set shot.

The video group was exposed to a set shot skill acquisition video. The information was delivered through a visual only sensory modality consisting exclusively of basketball shooting demonstration footage.

The audiovisual (video with narration) group received the same video content as the visual-only group but with the addition of voice-over narration. The video presented basketball shooting demonstration footage synchronized with verbal explanations of movement essentials. The audio content precisely matched the key movement components and was temporally aligned with the video presentation.

The group receiving video with narration and markers received the previous two information types plus additional visual markers. Research has demonstrated that incorporating guiding information considerably enhances retention test performance and increases learners' fixation duration toward and frequency of fixating on marked cue areas [47]. Additionally, research has indicated that physical marker cues (e.g., flashing arrows) in key information areas effectively guide attention [48]. On the basis of these findings, the guided dual-sensory video materials in this study incorporated localized markers targeting movement essentials, which appeared on corresponding limb segment displays without obscuring critical movement feature areas. The marking method employed low-saturation red flashing frames that appeared synchronously with movement essentials presented through both video and narration. Each marker was presented for 1 second and flashed only once before transitioning to a static red frame. Preliminary experimental results on changes in learning outcomes suggested that the presentation of the markers did not induce unintended visual bias, distract attention from critical movement features, or give rise to redundancy effects.

The filming protocol was established on the basis of the temporal sequence of set shot execution, incorporating the recommendations of expert consultations and referencing the basketball textbooks edited by Sun Minzhi in addition to findings in the literature [44]. The learning information provided in the video materials was consistent with both the Set Shot Movement Pattern Evaluation Scale and the Set Shot Motor Cognition Test, establishing alignment across learning materials, motor skill assessments, and motor cognition evaluations.

**Set shot motor cognition test.** The study assessed set shot motor cognition by using the Set Shot Motor Cognition Test, a multichoice questionnaire based on the Set Shot Movement Pattern Evaluation Scale. The test content aligned with the key movement points and essential elements highlighted in the instructional video. An initial draft of the test was reviewed by basketball professionals, and the draft was refined on the basis of their feedback. A pilot test was then conducted. In total, 50 valid questionnaire responses were received in the pilot test. The discrimination index (D) indicates item discriminatory ability and reliability, with higher values indicating greater discriminatory ability and reliability. Items with D values < 0.2 should be eliminated, and items with D > 0.4 have high discriminatory ability and reliability and should be retained [49]. The difficulty index (P) reflects item difficulty, with higher values indicating lower difficulty; the acceptable

range is from.20 to.80, with the optimal mean of.50 [49]. To enhance test validity, this study conducted a distractor analysis after the pilot test. Any distractors selected significantly more or less frequently than anticipated or chosen by less than 5% of the examinees were eliminated in subsequent refinements to the test. The Set Shot Motor Cognition Test was finalized by adjusting both the frequency and sequence of correct answers to ensure balanced distribution and randomization across items.

The test utilized the split-half method to assess internal consistency, with a reliability coefficient of 0.82, P values ranging from 0.39 to 0.89, and D values ranging between 0.22 and 0.83.

**Set shot movement pattern evaluation scale.** The measurement tools employed in this study exhibited high intercorrelations. The Set Shot Movement Pattern Evaluation Scale was developed primarily on the basis of key movement components in instructional videos, with additional reference to the literature and set shot learning materials [44]. A self-developed scale called the Set Shot Movement Pattern Evaluation Scale was used to evaluate movement pattern performance in upper-limb movement pattern, lower-limb movement pattern, and ball-handling movement pattern during the preparation and execution stages of a set shot. The preparation stage spans ball reception to movement initiation, whereas the execution stage extends from movement initiation until ball release [44].

The internal consistency of the Set Shot Movement Pattern Evaluation Scale items was determined on the basis of the pilot test data. On the basis of the valid sample ($N=50$), the total scale had a Cronbach's α coefficient of 0.88 [95% confidence interval: 0.86, 0.90], indicating excellent internal consistency [49].

Interrater reliability was assessed using videos of the first 10 participants and evaluated using Kendall's coefficient of concordance. The coefficients for upper-limb, lower-limb, and ball-handling movement patterns were 0.89, 0.90, and 0.94, respectively, with $\chi^2$ values of 21.20, 22.68, and 24.21, respectively; all values indicated significant concordance, implying good interrater reliability.

Internal reliability of raters was also assessed using videos of the top 10 participants, with 2 assessments being conducted 2 weeks apart to check test–retest reliability. Kendall's coefficient of concordance revealed high reliability, with coefficients of 0.93, 0.94, and 0.97 for upper-limb, lower-limb, and ball-handling movement patterns, respectively, and corresponding $\chi^2$ values of 25.10, 26.34, and 27.71. All values indicated significant consistency.

**Experimental procedure.** Expert interviews were conducted in which experts reviewed the experimental framework and tools and provided suggestions for modifications. A pilot test was then conducted to assess the reliability and validity of the research tools. Before the experiment commenced, the motor skills of the participants were evaluated; no between-group differences in motor skills were observed.

The main experiment involved the participants engaging in 3 weeks of learning, with two sessions per week (The experiment took place from September 9th, 2024 to September 27th, 2024.).

All three participant groups were instructed by the same physical education teacher; two assistants (graduate students in sports science) assisted with data collection. The procedures of the experimental phase were as follows. During the first session, pretest data on set shot movement result performance were collected by evaluating set shot movement pattern ratings (Set Shot Movement Pattern Evaluation Scale). Sessions 2–6 focused on motor skill acquisition, with each session comprising the following standardized procedures: (1) group warm-up exercises for all participants in the same condition group led by the physical education teacher (7–8 minutes); (2) teacher-directed, condition-specific instructional video viewing (repeated 3 times, 4–5 minutes); (3) guided practice with outcome-based feedback (teacher comments on movement accuracy alone without making corrections), maintaining equal feedback frequency across classes (two instances per participant; 10–12 minutes); (4) repeat video viewing (4–5 minutes); (5) repeat guided practice (10–12 minutes); (6) instruction to avoid practice outside of instructional hours. Immediately after the conclusion of the sixth session, assessments of set shot motor cognition (Set Shot Motor Cognition Test), set shot movement result performance, and set shot movement pattern ratings (Set Shot Movement Pattern Evaluation Scale) were administered. A retention test was administered 48 hours after the posttest (Fig 1).

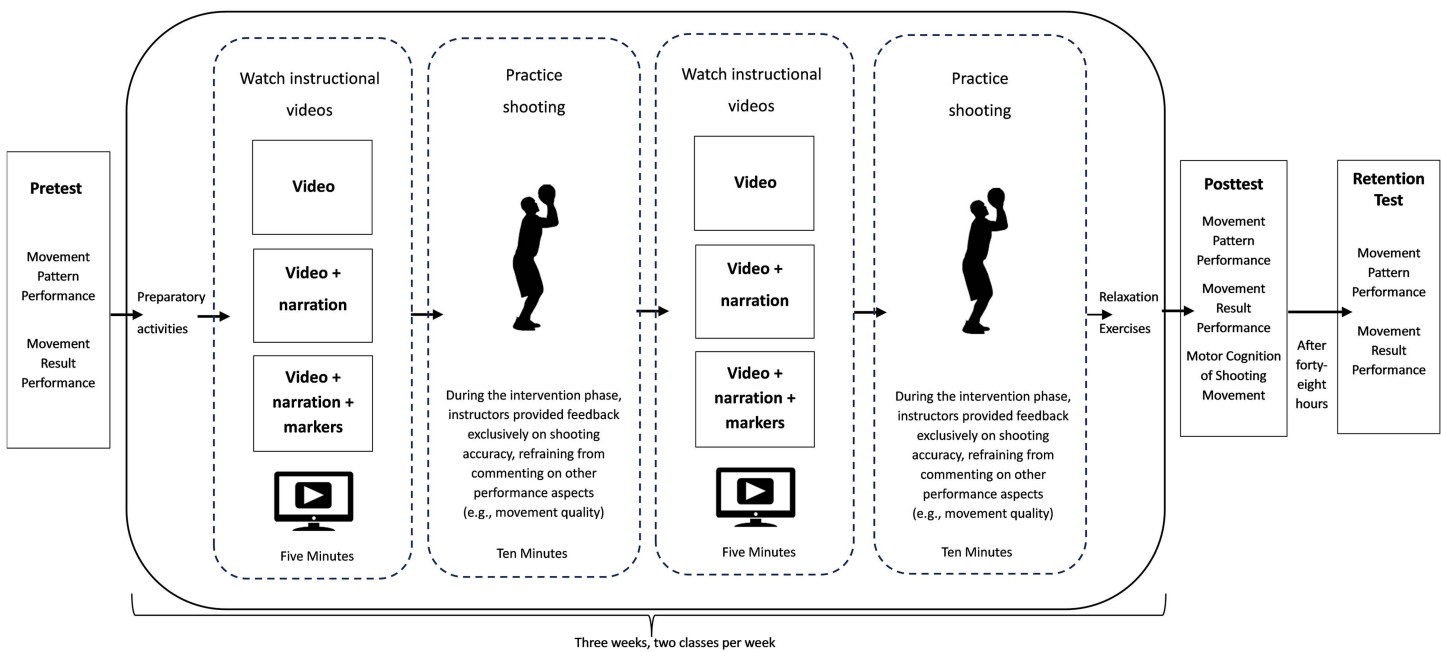

**Fig 1. Instructional Flowchart.**

## Learning outcome data collection

**Motor cognition.** Motor cognition was assessed using the self-developed Set Shot Motor Cognition Test, which comprised 18 multichoice questions. Each correct answer received 1 point, with a maximum score of 18 points for all correct answers and a minimum score of 0 points for all incorrect answers.

For example, the key point: In the preparation stage, the ball should be held between the chin and the chest. This is included as part of the multimedia teaching materials during instruction. It also appears as a single-choice question in the Set Shot Motor Cognition Test (e.g., "During the preparation stage, where should the ball be held?"). In the Movement Pattern Performance assessment, this is a scoring criterion. Expert raters observe the students' movements and evaluate them based on the key technical points.

**Movement result performance.** Movement result performance was assessed by asking the participants to apply the set shot method from a distance of 3 m from the center of the hoop. Participants were asked to take 10 shots. The scoring in this assessment was as follows: 0 points if the ball did not enter the hoop and did not touch the rim; 1 point if the ball touched the rim but did not go into the hoop; and 2 points if the ball went into the hoop. The total score for the 10 shots was used as the performance outcome score.

**Movement pattern performance.** Three experts evaluated movement pattern performance by using the Set Shot Movement Pattern Evaluation Scale. The final scores were determined by averaging the ratings of the three experts. The expert raters were blinded to the group assignment information during the evaluation process. Participants received 1 point for each element completed in accordance with the movement pattern criteria and 0 points for partial compliance or noncompliance. The total score represented the participant's adherence to the set shot movement pattern, with higher scores indicating better adherence. Participants received an overall movement pattern score (sum of scores from each part) and individual scores for each movement segment.

**Data processing.** Statistical analyses were performed in SPSS (Statistical Product and Service Solutions) version 27.0.

Before the experiment, the researcher conducted a homogeneity test for the three groups of participants to assess their set shot movement pattern performance and movement result performance. Kendall's coefficient of concordance was used to assess interrater and intrarater reliability, with the reliability threshold set at 0.80.

One-way ANOVA was used to evaluate between-group differences in motor cognition. If significant differences were identified, Scheffe's method was used for post hoc comparisons, with the significance level set at $\alpha = 0.05$.

A mixed-design ANOVA was employed to examine differences in set shot movement pattern performance and set shot outcome performance across the three information presentation groups (video, video with narration, and video with narration and markers) between the pretest and posttest. When significant group × time interactions were observed, simple main effects analyses were conducted, followed by post hoc comparisons using the least significant difference (LSD) method ($\alpha = .05$). When nonsignificant interactions were identified, the main effects were analyzed separately using LSD post hoc tests ($\alpha = .05$).

## Results

### Motor performance: Movement pattern performance

To investigate the effects of different information presentation methods on set shot movement pattern performance, this study examined the well-defined spatiotemporal structure of the set shot, which can be clearly divided into preparation and execution stages and involves control of the upper-limb, the lower-limb, and the ball-handling in three distinct segments. The pretest data revealed no significant differences in movement pattern performance total scores ($F = .051$, $p > .05$, $\eta^2 = .001$), verifying baseline homogeneity across the three learner groups. Table 1 presents a summary of the movement pattern performance scores before and after the intervention.

Regarding the upper-limb scores during the preparation stage of the movement patterns, the results revealed a significant time × group interaction ($F = 4.098$, $p < .05$, $\eta^2 = .060$). The results of a simple main effects analysis revealed that the group receiving video with narration and markers exhibited significantly superior performance at the posttest and retention test compared with the pretest ($F = 22.832$, $p < .05$, $\eta^2 = .347$); no significant difference was observed between the posttest and the retention test. The analysis results also indicated significant differences among the three groups at both the posttest and retention test ($F = 9.801$, $p < .05$, $\eta^2 = .132$; $F = 9.668$, $p < .05$, $\eta^2 = .130$). Post hoc comparisons revealed that the group receiving video with narration and markers scored significantly higher in upper-limb movement pattern preparation stage scores on the posttest and retention test than both the group receiving video information only and the group receiving audiovisual information ($p < .05$) did; no significant differences were observed between the video and audiovisual groups.

Regarding the lower-limb scores during the preparation stage of the movement patterns test, the analysis results revealed a nonsignificant time × group interaction effect ($F = 0.587$, $p = .645$, $\eta^2 = .009$). Neither the main effect of time ($F = 1.192$, $p = .305$, $\eta^2 = .009$) nor the main effect of group ($F = 0.009$, $p = .906$, $\eta^2 = .002$) reached significance.

The analysis results of ball-handling scores during the movement pattern preparation stage revealed a nonsignificant time × group interaction effect ($F = 1.374$, $p = .249$, $\eta^2 = .021$); however, a significant main effect of time was observed ($F = 4.546$, $p < .05$, $\eta^2 = .034$), with LSD post hoc test results revealing significantly higher posttest scores than pretest scores ($p < .05$). No significant differences were observed between the posttest and retention test or between the retention test and the pretest; the main effect of group was nonsignificant ($F = 1.850$, $p = .161$, $\eta^2 = .028$).

Regarding the total scores during the preparation stage of the movement patterns test, the results revealed a significant time × group interaction effect ($F = 3.817$, $p < .05$, $\eta^2 = .056$). The results of a simple main effects analysis also revealed significant differences across time points for the group receiving video with narration and markers ($F = 21.860$, $p < .05$, $\eta^2 = .337$), with LSD post hoc test results uncovering the following progression in preparation stage total scores: posttest>retention test>pretest (all $p < .05$). Regarding group differences, significant variations were observed among the

 PLOS One

**Table 1. Summary of means and standard deviations of basketball shooting movement pattern performance scores.**

| | | | Video (n = 44) | Video with narration (n = 44) | Video with narration and markers (n = 44) |
|---|---|---|---|---|---|
| | | | M ± SD | | |
| Pre-test | Preparation stage | Upper-limb | 1.43 ± 0.625 | 1.68 ± 0.561 | 1.61 ± 0.754 |
| | | Lower-limb | 1.43 ± 0.395 | 1.43 ± 0.473 | 1.41 ± 0.693 |
| | | Ball-handling | 0.56 ± 0.490 | 0.57 ± 0.482 | 0.56 ± 0.434 |
| | | Preparation stage total score | 3.42 ± 1.510 | 3.41 ± 1.765 | 3.43 ± 1.619 |
| | Execution stage | Upper-limb | 1.34 ± 1.238 | 1.48 ± 1.372 | 1.34 ± 1.256 |
| | | Lower-limb | 0.93 ± 0.398 | 0.94 ± 0.371 | 0.95 ± 0.248 |
| | | Ball-handling | 0.43 ± 0.501 | 0.43 ± 0.501 | 0.45 ± 0.504 |
| | | Execution stage total score | 2.68 ± 1.520 | 2.84 ± 1.765 | 2.73 ± 1.619 |
| | Shot Movement Pattern Performance total score | | 6.20 ± 2.226 | 6.32 ± 2.122 | 6.18 ± 2.072 |
| Post-test | Preparation stage | Upper-limb | 1.68 ± 0.740 | 1.75 ± 0.781 | 2.27 ± 0.499 |
| | | Lower-limb | 1.43 ± 0.625 | 1.52 ± 0.698 | 1.61 ± 0.579 |
| | | Ball-handling | 0.57 ± 0.501 | 0.73 ± 0.451 | 0.84 ± 0.370 |
| | | Preparation stage total score | 3.68 ± 1.253 | 3.98 ± 1.355 | 4.73 ± 0.949 |
| | Execution stage | Upper-limb | 1.73 ± 1.370 | 1.98 ± 1.438 | 2.32 ± 1.006 |
| | | Lower-limb | 0.89 ± 0.443 | 0.97 ± 0.283 | 0.95 ± 0.211 |
| | | Ball-handling | 0.45 ± 0.504 | 0.59 ± 0.497 | 0.57 ± 0.501 |
| | | Execution stage total score | 3.05 ± 1.855 | 3.57 ± 1.744 | 3.77 ± 1.344 |
| | Shot Movement Pattern Performance total score | | 6.73 ± 2.591 | 7.55 ± 2.680 | 8.50 ± 1.911 |
| Retention Test | Preparation stage | Upper-limb | 1.61 ± 0.841 | 1.66 ± 0.745 | 2.20 ± 0.462 |
| | | Lower-limb | 1.48 ± 0.731 | 1.50 ± 0.699 | 1.45 ± 0.548 |
| | | Ball-handling | 0.57 ± 0.466 | 0.66 ± 0.479 | 0.73 ± 0.451 |
| | | Preparation stage total score | 3.64 ± 1.331 | 3.82 ± 1.451 | 4.39 ± 0.970 |
| | Execution stage | Upper-limb | 1.50 ± 1.422 | 2.00 ± 1.276 | 2.09 ± 0.884 |
| | | Lower-limb | 0.95 ± 0.371 | 0.98 ± 0.245 | 0.97 ± 0.302 |
| | | Ball-handling | 0.45 ± 0.504 | 0.57 ± 0.501 | 0.43 ± 0.501 |
| | | Execution stage total score | 2.89 ± 1.794 | 3.57 ± 1.634 | 3.48 ± 1.110 |
| | Shot Movement Pattern Performance total score | | 6.52 ± 2.328 | 7.39 ± 2.599 | 7.84 ± 1.725 |

three groups on both the posttest ($F = 8.900$, $p < .05$, $\eta^2 = .121$) and retention test ($F = 4.914$, $p < .05$, $\eta^2 = .061$). Post hoc comparison results indicated that the group receiving video with narration and markers significantly outperformed both the video and audiovisual groups in total movement pattern scores during the posttest and retention test ($p < .05$), although no significant difference was observed between the video and audiovisual groups on these tests.

The results of the analysis of upper-limb scores during the movement execution stage indicated the absence of a significant time × group interaction effect ($F = 1.545$, $p = .190$, $\eta^2 = .023$); however, a significant main effect of time was observed ($F = 14.141$, $p < .05$, $\eta^2 = .099$), with LSD post hoc test results revealing that both posttest and retention test scores were significantly higher than the pretest scores ($p < .05$). No significant difference was observed between the posttest and the retention test, and the main effect of group was nonsignificant ($F = 1.966$, $p = .144$, $\eta^2 = .030$).

The results of the analysis of the lower-limb scores during the movement execution stage indicated the absence of a significant time × group interaction effect ($F = 0.330$, $p = .813$, $\eta^2 = .005$); neither the main effect of time ($F = 1.909$, $p = .169$, $\eta^2 = .015$) nor the main effect of group ($F = 1.122$, $p = .329$, $\eta^2 = .017$) reached significance.

The analysis results of ball-handling scores during the movement execution stage indicated the absence of a significant time × group interaction effect (F = 0.745, p = .551, η² = .011); additionally, neither the main effect of time (F = 1.930, p = .147, η² = .015) nor the main effect of group (F = 0.544, p = .582, η² = .008) reached significance.

The analysis results of total scores during the movement execution stage revealed no significant time × group interaction effect (F = 1.058, p = .371, η² = .016); nevertheless, a significant main effect of time was observed (F = 11.484, p < .05, η² = .082), with LSD post hoc test results indicating that the posttest scores were significantly higher than the pretest scores (p < .05), the retention test scores were significantly higher than the pretest scores (p < .05), and no significant difference existed between the posttest and the retention test. Additionally, the main effect of group was nonsignificant (F = 1.937, p = .148, η² = .029).

The results of an analysis of total movement pattern performance scores revealed a significant time × group interaction effect (F = 3.228, p < .05, η² = .048). Additionally, the results of a simple main effects analysis revealed the following: the audiovisual group exhibited significant differences with respect to time between the pretest and subsequent tests (F = 6.119, p < .05, η² = .125), with LSD post hoc test results indicating that both the posttest and retention test scores were significantly higher than those of the pretest (both p < .05), although no significant difference was noted between the posttest and retention test. The group receiving video with narration and markers also exhibited significant differences with respect to time (F = 25.056, p < .05, η² = .374), with the order of the LSD test results being posttest > retention test > pretest (all p < .05). Significant differences were also observed among the three groups at both the posttest (F = 5.921, p < .05, η² = .084) and retention test (F = 3.907, p < .05, η² = .057). Post hoc comparison results revealed that the group receiving video with narration and markers significantly outperformed the video group on both the posttest and retention test (p < .05). No other between-group differences reached significance.

The aforementioned results indicate that different combinations of information presentation significantly affected the students' set shot movement pattern performance. Specifically, both the group receiving audiovisual instruction and the group receiving instruction through video with narration and markers were significantly associated with enhanced movement patterns after the intervention, with the group receiving video with narration and markers group scoring higher on the retention test than the video and audiovisual groups. Additionally, spatiotemporal dimension analysis results revealed that the learners receiving the guided dual-sensory instruction achieved the highest scores in upper-limb movement patterns during the preparation and total preparation stages.

**Motor performance: Movement result performance.** The pretest data indicated the absence of significant differences in movement result performance scores among the three groups (F = 0.047, p > .05, η² = .001), verifying baseline homogeneity. Table 2 presents a summary of the pre- and postintervention scores for movement result performance.

The results of an analysis of movement result-performance scores revealed a significant time × group interaction effect (F = 5.810, p < .05, η² = .083). Simple main effects analysis results revealed that the video group exhibited significant differences with respect to time (F = 7.762, p < .05, η² = .151), with LSD post hoc test results revealing posttest scores that were significantly higher than the pretest scores, (p < .05) and retention test scores that were significantly higher than pretest scores (p < .05). However, no significant difference was observed between the scores on the posttest and the retention test. Additionally, the

**Table 2. Summary of means and standard deviations of basketball shooting movement result performance scores.**

|  | Video (n = 44) | Video with narration (n = 44) | Video with narration and markers (n = 44) |
|---|---|---|---|
|  | **M ± SD** |  |  |
| **Pre-test** | 7.30 ± 4.825 | 7.45 ± 3.676 | 7.54 ± 4.043 |
| **Post-test** | 8.27 ± 5.137 | 9.52 ± 3.782 | 10.39 ± 4.065 |
| **Retention Test** | 7.70 ± 5.033 | 9.23 ± 3.839 | 9.41 ± 4.369 |

group receiving audiovisual instruction exhibited significant differences with respect to time (F = 22.274, $p < .05$, $\eta^2 = .346$), with LSD test results revealing posttest scores that were significantly higher than the pretest scores ($p < .05$) and retention test scores that were significantly higher than the pretest scores ($p < .05$); however, no significant difference was observed between the posttest and the retention test. Finally, the group receiving video with narration and markers exhibited significant differences with respect to time (F = 38.144, $p < .05$, $\eta^2 = .470$), with LSD comparison results revealing that posttest scores>retention test scores>pretest scores (all $p < .05$). Moreover, simple main effects analysis results uncovered no significant differences between groups (F = .047, $p = .954$, $\eta^2 = .001$; F = 2.606, $p = .078$, $\eta^2 = .039$; F = 1.955, $p = .146$, $\eta^2 = .029$).

These results indicate that the three instructional methods—visual-only (video), audiovisual (video with narration), and guided dual-sensory information (video with narration and markers)—all were associated with significantly increased movement result performance. Specifically, as reflected in both the posttest and retention test scores compared with the pretest scores, the students' shooting accuracy significantly increased. However, when in the analysis of between-group effects by information modality (single-sensory information group vs. dual-sensory information group vs. and guided dual-sensory information group), only the guided dual-sensory information group exhibited significantly increased movement result outcomes on the posttest compared with the visual-only group.

These results demonstrate that the three instructional methods of video, audiovisual, and guided dual-sensory instruction were all significantly associated with better motor performance. Specifically, learners' shooting accuracy considerably improved on both the posttest and retention test scores compared with the pretest. However, when comparing the between-group effects among the visual-only group, audiovisual group, and guided dual-sensory information group, only the guided dual-sensory instruction group exhibited substantially superior movement result outcomes than the video group on the posttest.

**Motor cognition.** One-way ANOVA was conducted to evaluate between-group differences in motor cognition for set shot actions. If significant differences were identified, post hoc comparisons were conducted using Scheffe's method, with significance set at $\alpha = 0.05$.

As indicated in Table 3, no significant differences were observed among the three groups in lower-limb motor cognition during the preparation stage, ball-handling cognition during the preparation stage, or lower-limb motor cognition during the execution stage. However, significant differences were noted in upper-limb motor cognition during the preparation stage (F = 6.565, $p < .05$, $\eta^2 = .089$), in total preparation stage cognition scores (F = 6.608, $p < .05$, $\eta^2 = .118$), in execution stage ball-handling cognition (F = 13.724, $p < .05$, $\eta^2 = .175$), and in total shooting cognition scores (F = 15.566, $p < .05$, $\eta^2 = .371$). Scheffe post hoc analysis results indicated that the guided dual-sensory instruction group and the video with narration group both significantly outperformed the video group on these measures. Regarding execution stage upper-limb motor cognition (F = 13.034, $p < .05$, $\eta^2 = .216$) and total execution stage cognition scores (F = 18.001, $p < .05$, $\eta^2 = .299$), the scores of the guided dual-sensory instruction group surpassed those of both the video with narration group and the video group. Similarly, the scores of the video with narration group for these measures also exceeded those of the video group (Table 3).

The results demonstrate that the modality of information presentation significantly affected the students' motor cognition during set shot performance. First, both the video with narration and video with narration plus markers groups scored significantly higher on motor cognition than the video group did after training. Second, the spatiotemporal dimension analysis results revealed that the dual-sensory information group (video with narration) exhibited superior upper-limb motor cognition, whereas the guided dual-sensory information group (video with narration plus markers) achieved the highest scores among all three groups for both upper-limb motor cognition and total scores during the execution stage.

## Discussion

This study investigated the effect of the method used to present information on basketball set shot movement learning outcomes. The study conducted a detailed analysis of motor performance and motor cognition across various movement stages and limb segments to evaluate the motor learning process.

 

**Table 3. Between-Group Differences in Motor Cognition of Shooting Movement.**

| | | Video (n = 44) | Video with narration (n = 44) | Video with narration and markers(n = 44) | F | p | η² |
|---|---|---|---|---|---|---|---|
| | | M ± SD | | | | | |
| **Preparation stage** | **Upper-limb motor cognition** | 2.48 ± 0.952[ac] | 3.20 ± 0.734[c] | 2.95 ± 1.140[a] | 6.565 | .002* | .089 |
| | **Lower-limb motor cognition** | 2.68 ± 0.771 | 2.91 ± 0.858 | 2.93 ± 0.925 | 1.154 | .319 | .036 |
| | **Ball-handling motor cognition** | 1.09 ± 0.520 | 1.32 ± 0.561 | 1.27 ± 0.544 | 2.165 | .119 | .019 |
| | **Preparation stage motor cognition total score** | 6.25 ± 1.416[ac] | 7.43 ± 1.576[c] | 7.16 ± 1.778[a] | 6.608 | .002* | .118 |
| **Execution stage** | **Upper-limb motor cognition** | 2.18 ± 1.084[ac] | 2.95 ± 1.584[bc] | 3.64 ± 1.296[ab] | 13.034 | .000* | .216 |
| | **Lower-limb motor cognition** | 0.59 ± 0.497 | 0.75 ± 0.438 | 0.68 ± 0.471 | 1.272 | .284 | .006 |
| | **Ball-handling motor cognition** | 0.73 ± 0.817[ac] | 1.27 ± 0.694[c] | 1.52 ± 0.664[a] | 13.724 | .000* | .175 |
| | **Execution stage motor cognition total score** | 3.50 ± 1.439[ac] | 4.98 ± 2.226[bc] | 5.84 ± 1.804[ab] | 18.001 | .000* | .299 |
| **Shot motor cognition total score** | | 9.75 ± 2.300[ac] | 12.41 ± 3.301[c] | 13.00 ± 3.035[a] | 15.566 | .000* | .371* |

*$p < 0.05$.

[a]Video with narration and markers group performed significantly better than did video group.

[b]Video with narration and markers group performed significantly better than did video with narration group.

[c]Video with narration group performed significantly better than did video group.

The three multimedia teaching methods all positively affected both motor performance and motor cognition, although the effects varied in magnitude. In analyzing the experimental data and comparing methods of presenting information across different movement stages and limb segments, this study discovered that learning outcomes differ with the method, highlighting a need for further exploration of this topic.

First, regarding movement pattern performance, a comparison before and after multimedia instruction revealed that visual–auditory yielded the best motor skills retention results. Despite some studies have suggested that multimedia instruction does not affect motor learning skills [20,30,31]; however, in the present study, set shot movement pattern performance significantly improved in the video with narration and video with narration and markers groups. Guided dual-sensory information aligns with both the dual-channel principle and the signaling principle of CLT [11]. Several studies have demonstrated that multimedia instructional environments lead to enhanced learning performance [3,27,50]. Research has demonstrated that consistent audiovisual stimuli can facilitate procedural perceptual-motor learning [51]. In one study, the simultaneous presentation of video, text, and music enabled learners to effectively observe and understand the detailed movements of correct badminton footwork and shot techniques [52]. The findings of one study comparing the effects of video, verbal, and self-learning instructions on golf swing performance revealed that although the self-learning group exhibited superior performance on the immediate posttest, both the video and verbal instruction groups outperformed the self-learning group on the 2-week delayed posttest, with the video instruction group exhibiting the most substantial improvement. These findings suggest that the effectiveness of video-based instruction may require time to manifest fully [53].

The results of an analysis of movement staging and limb segments revealed that the group receiving guided dual-sensory instruction achieved the optimal upper-limb movement pattern performance during the preparation stage. This result likely reflects superior absorption of preparatory upper-limb movement techniques from the combined video, narration, and visual markers. First, the marker information directed attention to critical details, enabling learners to establish recall procedures during movement execution and self-detect errors in their movements. Motor skill acquisition follows a

staged information processing flow comprising stimulus identification, response selection, and response programming. In the perceptual stage, learners receive feedback regarding environmental and bodily states through sensory channels that enables them to identify action goals and contextual cues. During the decision stage, learners retrieve movement schemas from long-term memory on the basis of perceptual input to determine and plan motor responses. The execution stage involves translating motor plans into neuromuscular commands to execute actions [38]. Because information input occurs during stimulus identification in the perceptual stage, learners must identify key information before interpreting its attributes. Second, shooting movements require learners to have sufficient opportunities to adjust movement parameters during the preparation stage. Because upper-limb and ball-handling actions also involve the visual field, learners position their body relative to the environment and the basket before making timely corrections to their movements using intrinsic sensory feedback when executing a shot [4,38]. These findings demonstrate that multimedia instruction in shooting improves the learner's skill. However, the literature has predominantly analyzed movements as complete units, neglecting the component–whole analysis required by the part-to-whole principle that is integral to sound shooting pedagogy.

The superior learning outcomes of the group receiving guided dual-sensory instruction suggest that these learners did not experience redundancy effects despite the increased information load during basketball shooting training. Research employing different multimedia presentation formats to teach shooting skills revealed that video and on-screen text produced the least optimal results in movement integration instruction. The researchers attributed this result to competition between two types of visual stimuli, particularly to the redundant information provided through text (i.e., the redundancy effect) [54]. However, when information resources are properly allocated, learning results can be optimized. In the current study, for basketball shooting, adding marker information did not create a redundancy effect; rather, it provided attentional guidance, resulting in superior movement patterns and shooting skills [47]. The redundancy effect involves a "cognitive efficiency penalty caused by information repetition" [5]. In this study, the video (visual stimulus) showed the relevant techniques in their entirety, the narration (auditory stimulus) explained the principles underlying the movements, and the markers (visual cues) directed the learner's attention to key areas; these three elements complemented one another rather than introducing redundancy effects. This finding lends support to the hypothesis that in motor skill instruction, when multimodal information elements perform distinct roles, they can mitigate redundancy and lead to the optimal allocation of cognitive resources.

Although this study did not employ eye-tracking or subjective attention scales to directly record learners' visual attention distribution, the signaling effect documented in analyses of multimedia learning and related empirical studies supports the effectiveness of marker cues in guiding attention. For example, Zhang et al. demonstrated that overlaying expert gaze trajectories on videos effectively guided learners to key areas and optimized visual search patterns [55]. On the basis of this evidence and the enhanced motor learning outcomes observed in the present study, this study hypothesized that visual marker cues likely enhanced learning by directing learners' visual attention to critical movement components. However, the potential roles of other attention-guiding mechanisms in this process warrant further study [56].

When analyzing movement limb segments, this study observed both positive gains and nonsignificant changes—particularly in the lower-limb movement patterns during the preparation and execution stages of shooting. This result may be attributable to (1) the low technical requirements to perform lower-limb actions (e.g., knee flexion or push-off); (2) seventh-graders' fundamental squatting movements having reached near-automatic, near adult-level neuromuscular control through motor experience [57], consistent with the principle that foundational skills mature earliest in child motor development. Consequently, the instructional interventions resulted in limited benefits to these well-established basic movements. Hence, no significant differences were observed in the lower-limb learning outcomes across the visual-only (video), audiovisual (video with narration), or guided dual-sensory (video with narration and markers) conditions. This finding also explains the inconsistencies observed in studies regarding the efficacy of multimedia in motor skill learning; these studies have neither employed standardized information presentation methods nor conducted time- or segment-specific analyses, limiting the accuracy and specificity of their results.

Second, pre–post instructional comparisons of movement result performance revealed considerable improvements in shooting accuracy across all three groups, with both posttest and retention test scores exceeding the scores on the pretest. When comparing movement pattern performance and movement result performance, the temporal trends in learning gains were consistent—regardless of information presentation method, six instructional sessions induced positive changes in set shot movement patterns that subsequently translated to enhanced movement results. Between-group analysis results revealed that the group receiving guided dual-sensory information exhibited considerably superior performance to that of the visual-only group during the posttest, verifying the superior effectiveness of augmented attention-guiding information at this stage. This finding supports the dual coding theory, which suggests that learning with both visual and auditory information is more effective than learning with one form is [18]. Studies have reported that combining visual information with auditory instructions is more effective than combining visual information with textual instructions [6,58]. Nevertheless, the retention test results revealed no significant differences in shooting accuracy (movement result performance) among the three groups, primarily because the group receiving guided dual-sensory information group exhibited substantial performance reductions over time. Although dual-sensory guidance temporarily enhanced movement pattern performance, its learning effects may be unstable and exhibit noticeable performance decay during retention testing. This finding suggests that the intervention may have facilitated only the shallow encoding of motor skills. This finding aligns with those of a study [54] indicating that neither multimedia presentation method nor measurement phase substantially enhanced shooting accuracy retention.

Motor learning is the process through which individuals achieve persistent changes in motor skills through accumulated practice and experience [4]. Although the guided dual-sensory learning group exhibited considerably superior movement pattern performance than the visual-only group, this advantage did not consistently translate into greater shooting accuracy. This result indicates that although information presentation substantially influenced movement pattern learning, shooting accuracy—a higher-order motor skill —likely requires multisystem integration and thus demands more prolonged, specialized practice to achieve substantial improvements in precision.

According to Fitts' motor learning theory, motor skill learning occurs in three stages (hierarchical skill model) [39]. The first stage is the cognitive stage, during which learners learn which actions to perform. This stage is brief but demanding [4]. The second stage is the fixation stage, during which learners focus on task-specific information and refine their movements, which remain unstable and require external feedback. This stage requires less effort but lasts longer [59]. The third stage is the autonomous stage, during which movements become automatic, effortless, accurate, and efficient [60–62]. The participants were engaged in the earliest stages of shooting learning, in which movement execution is slow and unsteady and improved movement patterns do not translate to increased shooting accuracy, reflecting a form–function dissociation in complex motor skills that is consistent with the predictions of the hierarchical skill acquisition model [39]. More importantly, movement result performance alone cannot evaluate the temporal and spatial process of learners' movements. Outcome performance alone does not fully reflect the effectiveness of the learning process because it does not capture the timing and spatial aspects of movements. This may explain why findings have often varied in motor learning studies.

Third, the current study revealed significant differences in basketball shooting cognition across the three investigated methods of presenting information. The video with narration and video with narration and markers groups exhibited better cognitive performance after the intervention compared with that of the video group, and no significant differences were observed between the video with narration and video with narration and markers groups. According to multimedia learning cognitive theory, visual and auditory materials are processed through separate channels, with each having a limited capacity [17]. In ths study, using narration instead of text prevented competition in the visual channel, distributing the cognitive load and reducing overload [17]. Research showed that multisensory stimulation not only facilitates early perceptual processing and motor responses but also enhances higher cognitive processes such as information discrimination and response selection [63]. Adding text to video did not improve cognitive scores among the basketball learners [54],

suggesting that combining images and text leads to visual channel overload. This study analyzes the motor cognitive of basketball set shot and identifies phenomena worth discussing. When examined through the lens of the spatiotemporal dimensions of movement staging, the dual-sensory information presentation was highly effective in promoting upper-limb motor cognition, particularly during the execution stage, during which the group receiving video with narration and markers (guided dual-sensory information) significantly outperformed the group receiving video with narration (audiovisual information), whereas the group receiving a video (visual-only information) exhibited the poorest performance. These results may be attributable to differences in information presentation, specifically the attention-guiding effect of marker cues. This study hypothesized that guiding learners' visual attention to key movement components through markers would reduce information search costs and enhance motor cognition. This difference is likely related to markers guiding attention, which influences cognitive processing and learning outcomes [6]. This finding aligns with those of other studies on attention guidance. In basketball learning, attention guidance has been demonstrated to reduce cognitive load and improve visual search accuracy [33,64]. From the perspective of the external load, reducing the dispersion of attention can help learners more effectively allocate focus. Additionally, the intrinsic load is reduced because the amount of content that must be simultaneously processed is lower. Because the participants in the present study were beginners, they might have struggled to focus on key information without attention guidance [65]. This would align with the findings of studies comparing visual attention in expert athletes and beginners [66]. Regarding motor cognition, no significant differences were observed in lower-limb cognition scores among the three groups during either the preparation or execution stages, a result consistent with findings in the literature on movement pattern performance. This result suggests a correlation between motor cognition and motor performance and reflects the requirement to consider cognitive differences in movement staging and motor sequencing when complex technical skills are taught through multimedia, particularly when the learned actions involve multiple limb segments.

The findings of this study validate the guided dual-sensory information strategy based on the dual-channel processing and signaling principles of the CTML. The findings of this study also demonstrate that (1) the dual-sensory information presentation considerably enhanced learners' movement pattern performance, with the addition of guiding information also enhancing movement pattern retention and optimizing upper-limb movement patterns during the preparation stage; (2) all three groups exhibited improved movement result performance after training, with the group receiving guided dual-sensory instruction exhibiting substantially superior movement results to those of the video group on the posttest; (3) both the audiovisual group and guided dual-sensory instruction group exhibited significant improvements in motor cognition after learning, with learners achieving the optimal upper-limb motor cognition during the execution stage when receiving guided dual-sensory information.

In this study, spatiotemporal analysis revealed that optimal presentation methods can enhance motor learning. The present study used spatiotemporal dimensions to examine learning progress, offering a novel perspective in assessing learning outcomes. The findings of this study have practical implications for physical education; they may guide teachers in providing learners with appropriate and effective multimedia learning materials. Instructional content regarding the principles of motor learning should incorporate movement staging and limb segments with distinct combinations of methods for presenting information. For example, in teaching set shots, using video with narration and markers (guided dual-sensory information) during the preparation stage can assist with studying upper-limb movement techniques, whereas video with narration (dual-sensory information) can assist with the execution stage motor learning. Furthermore, in learning lower-limb movements, information need only be presented using a video (a single-modality format). Employing such a tailored approach can enhance motor learning and support digital transformation in physical education.

This study has several limitations. The study included a limited number of middle school students; therefore, the results should be generalized with caution. Additionally, the findings indicating that marker cues enhance learning outcomes through attention guidance mechanisms were derived indirectly in the absence of direct attentional measurement. Future

research should explore different learning content and include a larger, more diverse cohort of learners across various sports disciplines. Future studies should incorporate expert learners to examine the differential effects of information presentation on individuals who have already acquired fundamental motor skills.

## Conclusion

Providing guided dual-sensory information can effectively enhance learning outcomes in basketball shooting. From a spatiotemporal perspective, no single method of presenting information can optimize learning across all stages of a set shot. The results of an in-depth spatiotemporal analysis indicated that the application of guided dual-sensory information yielded the optimal motor learning outcomes during the movement pattern preparation stage, with required movement guidance of the upper-limb segments being greater than that for other body segments. Future research should further explore how different methods of presenting information affect motor learning in digital physical education, with emphasis placed on examining the learning process through a spatiotemporal movement lens. Additionally, research should investigate the application of guided dual-sensory information tailored to the unique characteristics of movements in different sports. Enhancing the design of learning materials for physical education can provide support for the implementation of digital education.

## Acknowledgments

Special thanks to National Tsing Hua University in Taiwan support.

## Author contributions

**Conceptualization:** Liwa Sha, Wen Hsin Chiu.

**Funding acquisition:** Liwa Sha, Wen Hsin Chiu.

**Methodology:** Liwa Sha, Wen Hsin Chiu.

**Project administration:** Wen Hsin Chiu.

**Supervision:** Liwa Sha.

**Validation:** Liwa Sha, Wen Hsin Chiu.

**Writing – original draft:** Liwa Sha, Wen Hsin Chiu.

**Writing – review & editing:** Liwa Sha, Wen Hsin Chiu.

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
