## [Decision Letter · Decision Letter 0]

20 Jul 2025

Dear Dr. Chiu,

Thank you for submitting your manuscript to PLOS ONE. After careful consideration, we feel that it has merit but does not fully meet PLOS ONE’s publication criteria as it currently stands. Therefore, we invite you to submit a revised version of the manuscript that addresses the points raised during the review process.

Thanks for your submission to PLOS One.  This manuscript reports a timely and interesting study of how unimodal and multimodal instructional methods affect motor learning of the basketball set shot for middle school students. While the study was rigorously conducted, the manuscript requires major revisions before it can be published.  Specifically, the Introduction would benefit from clearer writing and greater theoretical integration and more explicit explanation of why the set shot is well suited for revealing the benefits of multisensory learning.  Moreover, the Method section would benefit from additional details concerning participant demographics, randomization, and blinding. Finally, both reviewers point out that a repeated measures ANOVA is better suited for analyzing the pre- and post-test data than separate ANOVAs for each test.  I encourage the authors to revise the manuscript in response to these criticisms, as well as additional detailed points raised by R1 and R2, and I will attempt to recruit the same reviewers to evaluate the extent to which their feedback is addressed sufficiently.

We look forward to receiving your revised manuscript.

Kind regards,

Laura Morett

Academic Editor

PLOS ONE

Journal Requirements:

“Supported by the Independent Innovation Fund of Jilin Sport University”

“Supported by the Independent Innovation Fund of Jilin Sport University”

4. In the online submission form, you indicated that [Insert text from online submission form here].

6. Please ensure that you include a title page within your main document. You should list all authors and all affiliations as per our author instructions and clearly indicate the corresponding author.

Reviewers' comments:

Reviewer's Responses to Questions

**Comments to the Author**

1. Is the manuscript technically sound, and do the data support the conclusions?

Reviewer #1: Partly

Reviewer #2: Yes

2. Has the statistical analysis been performed appropriately and rigorously?

Reviewer #1: I Don't Know

Reviewer #2: No

3. Have the authors made all data underlying the findings in their manuscript fully available?

Reviewer #1: Yes

Reviewer #2: Yes

4. Is the manuscript presented in an intelligible fashion and written in standard English?

Reviewer #1: Yes

Reviewer #2: Yes

Reviewer #1: The manuscript needs to be revised with regard to the presentation of methodology and results.

In terms of statiqtic treatments, it would appear that there was one anova performed pre-test and one post-test.

It would be more relevant to carry out a repeated-measure anova (pre/post/retention) * intervention group

For other comments, please see the attached document.

Reviewer #2: This manuscript investigates how different multimedia instructional methods (specifically visual only, visual plus auditory, and guided dual sensory using visual and auditory information with markers) affect motor learning of the basketball set shot among middle school students. The authors use a spatiotemporal analysis framework to evaluate both movement pattern performance and motor cognition across different stages of movement and various limb segments. The study is timely and addresses an important question in physical education and motor learning: how can digital instructional tools be optimized for teaching complex motor skills?

The study is methodologically sound, with a clear experimental design, a substantial sample size (N = 132), and careful measurement of both cognitive and physical aspects of motor learning. The use of retention testing and inter/intra-rater reliability measures further strengthens the rigor of the work. The application of a spatiotemporal lens to instructional effectiveness is a novel and meaningful contribution to the field.

That said, I believe the manuscript requires major revisions prior to consideration for publication. The manuscript would benefit from clearer writing, greater theoretical integration with the presented motor learning frameworks, and a deeper discussion of findings, limitations and generalizability.

Major Concerns:

1. Lack of a cohesive theoretical framework in the Introduction

While the authors reference several relevant learning theories (e.g., dual coding theory, cognitive load theory, multimedia learning), these frameworks are introduced in a fragmented way. The Introduction feels like a collection of loosely connected concepts rather than a synthesized argument leading to the present study. As a result, it is unclear how the theoretical background specifically motivates the study design or outcome measures.

In addition, a clear rationale for why the basketball set shot was chosen as the target skill is missing. Although it is noted as a common scoring technique, the authors do not explain why this particular skill is ideally suited for examining dual-sensory instructional methods or spatiotemporal learning. For example, is the set shot particularly reliant on visual modeling? Does it break down neatly into movement stages and limb segments more than other skills would? Establishing this would strengthen the case for generalizability and relevance.

Lastly, a central hypothesis or set of testable predictions is not explicitly stated. This makes it difficult to discern what the authors expected and how the results support or challenge their assumptions.

2. Insufficient methodological details

The study does not provide adequate information about participant demographics beyond age (e.g., gender distribution, physical literacy or motor skill baseline beyond set shot experience). This is especially important given that sex or experience level could potentially influence motor learning or responsiveness to different instructional modalities.

Additionally, the randomization process is not described in sufficient detail. The authors state that participants were randomly assigned to one of three instructional groups, but there is no mention of how this was done (e.g., computer-generated random numbers, stratified sampling, class-based assignment). This limits confidence in the equivalence of groups at baseline.

It is also unclear whether the expert raters were blinded to group allocation during performance assessment. Without blinding, there is risk of observer bias, particularly since scoring was based on qualitative movement ratings.

3. Ambiguity in describing the guided “marker” cues

The “guided dual sensory” condition (i.e. the addition of visual markers) is insufficiently described. The manuscript does not clarify what the markers looked like (e.g., flashing frames, arrows, highlights?), where on the screen they appeared (e.g., limb segments, ball), how long they were displayed, or whether they were dynamic (moving) or static. These features are critical to understanding how attention might have been guided.

Moreover, there is no discussion of whether the markers might have introduced unintended visual biases, distracted from important movement features, or created a redundancy effect. Given that attentional guidance is central to the study’s rationale, this lack of detail undermines interpretability and reproducibility.

4. Statistical concerns

The authors do not report effect sizes, limiting interpretation of the practical significance of the findings. Additionally, the use of separate one-way ANOVAs may not be appropriate given the within-subject dependency across time. A mixed-effects ANOVA would better account for within-subject variation across timepoints and reduce the risk of Type I error from multiple comparisons. For example: group (between) × time (within).

5. Lack of discussion of null results

Several analyses yielded non-significant findings (e.g., between-group differences in posttest accuracy, lower-limb performance), but these are not addressed in the discussion. Ignoring these results limits the reader’s understanding of the full pattern of findings and weakens the credibility of the interpretations. A balanced discussion should acknowledge where the instructional interventions did not yield clear benefits.

6. Discrepancy between form improvement and performance outcome

Although the study shows improvements in movement patterns, these did not consistently translate into better shooting accuracy. The discussion does not address this discrepancy. A possible explanation may lie in the early stage of motor learning, where improved form does not yet yield measurable gains in performance outcomes. However, this should be explicitly discussed to clarify the practical implications of the findings.

7. Interpretations about attention without direct measurement

The discussion relies heavily on attentional guidance theory to explain the observed benefits of marker-based instruction. However, attention was not directly measured in the study (e.g., through gaze tracking or reported focus), making these interpretations speculative. This limitation should be acknowledged more explicitly, especially since attention is central to the study’s proposed mechanism.

Minor Revisions:

Minor revisions are recommended to improve grammar, sentence structure, and overall flow, as several sections contain typographical errors, awkward phrasing, and inconsistent formatting that could hinder readability. See some examples below:

1. Line 34: Verb tense should say ‘enables’

2. Line 164: missing space after the period

3. Line 212: Sentence is the past tense.

4. Line 455 – sentence ends with an unnecessary ‘is’

5. Line 511 – likely a typo ‘studing’ – ‘studying’

**Do you want your identity to be public for this peer review?** For information about this choice, including consent withdrawal, please see our Privacy Policy

Reviewer #1: No

Reviewer #2: No

---

## [Author Response · Author response to Decision Letter 1]

7 Sep 2025

Reviewer 1

1. Lack of a cohesive theoretical framework in the Introduction

1.1 While the authors reference several relevant learning theories (e.g., dual coding theory, cognitive load theory, multimedia learning), these frameworks are introduced in a fragmented way. The Introduction feels like a collection of loosely connected concepts rather than a synthesized argument leading to the present study. As a result, it is unclear how the theoretical background specifically motivates the study design or outcome measures.

We sincerely appreciate your insightful comments regarding the insufficient integration of theoretical frameworks in our manuscript. Your observation about "fragmented theoretical introduction leading to logical discontinuity" was particularly enlightening. We have systematically restructured the theoretical framework using the Cognitive Theory of Multimedia Learning (CTML) as the unifying thread (specific revisions detailed below). Should any deficiencies remain, we warmly welcome further guidance.

I. Explicitly established CTML as the theoretical foundation in the Introduction, with its dual-channel processing and signaling principles directly supporting the proposed "guiding dual-sensory information" research strategy.

The present study used Mayer’s cognitive theory of multimedia learning (CTML) as its framework. This theory posits that when learners receive information through properly designed visual and auditory channels, this information reduces extraneous cognitive load, promotes deep processing, and enhances motor learning outcomes [5]. On the basis of the dual-channel signaling principle of the CTML [6], this study proposed a visual + audio + markers guided information presentation strategy.�Lines�37-42

II. The Introduction now presents a theoretically grounded derivation process through Cognitive Load Theory (CLT) and Dual Coding Theory, forming a three-tiered "CTML-CLT-Dual Coding Theory" progressive logic.

Cognitive load theory focuses on how an individual allocates mental resources and suggests that insufficiently accounting for memory limits in instructional design can hinder learning [15,16]. Mayer combined cognitive load theory with dual code theory, creating the Cognitive Theory of Multimedia Learning [17]. Dual code theory suggests that individuals process information through both images and language; that is, learners receive instructional information through different sensory channels [18]. Multimedia instructional design should reduce unnecessary cognitive load and promote understanding of material [5].�Lines�69-74

III. Incorporating the specificity of motor learning, we employed CTML to clarify the information requirements for action execution, thereby optimizing the delivery methods of multimedia learning information, the scientific validity of content, and the operational appropriateness of movement staging.

Motor learning requires simultaneously processing multichannel information and executing physical movements; these processes compete for limited resources in the learner’s working memory [15, 32].�Lines�100 -104

The aforementioned findings highlight the need for carefully designed and organized motor learning materials to ensure effective information processing and optimal performance.�Lines�106-108

Reducing cognitive load during motor learning requires providing information to guide learner attention and facilitate the construction of mental models [15].�Lines�121-122

IV. When addressing current research limitations, we introduced attention-guiding design requirements based on CLT and CTML's dual-channel/signaling principles. The motor schema theory was proposed to establish foundations for analyzing shooting learning outcomes through spatiotemporal dimensions. Motor learning theories provided the analytical framework for precise assessment of instructional information's impact on participants' motor performance and cognitive representation via quantitative analysis of spatiotemporal data.

Cognitive load theory and the dual-channel with signaling principle of the CTML suggest that motor learning requires matching sensory channel capacity with attention-guiding design [6, 15].�Lines�133-135

Additionally, Schmidt’s schema theory suggests that phased practice establishes refined motor schemata and enhances movement adaptability [40].�Lines�144-146

Supplementary references

40. Schmidt RA. A schema theory of discrete motor skill learning. Psychol Rev. 1975;82(4): 225–260. doi: 10.1037/h0076770

The set shot is a highly visual motor skill [44] that requires learners to establish internal representations by observing the movements, a process consistent with the fundamental assumptions of observational learning and imitation in multimedia learning theory.�Lines�167-170

V. Both research question derivation and Discussion sections strictly align with theoretical presuppositions. The results respectively validate relevant theories, echoing the problem formulation and inferences in the Introduction to ensure theoretical consistency from hypotheses to conclusions.

This study hypothesized that incorporating verbal narration and visual markers (red flashing frames) in instructional videos would enhance learners’ focus on critical motor learning information, increasing movement pattern performance and motor cognition outcomes. This design is consistent with the core proposition of CLT, which posits that learners most effectively manage limited cognitive resources by focusing on the most task-relevant information.�Lines�189-194

Guided dual-sensory information aligns with both the dual-channel principle and the signaling principle of CLT [11]. �Lines�606-607

The findings of this study validate the guided dual-sensory information strategy based on the dual-channel processing and signaling principles of the CTML. �Lines�767-768

1.2 In addition, a clear rationale for why the basketball set shot was chosen as the target skill is missing. Although it is noted as a common scoring technique, the authors do not explain why this particular skill is ideally suited for examining dual-sensory instructional methods or spatiotemporal learning. For example, is the set shot particularly reliant on visual modeling? Does it break down neatly into movement stages and limb segments more than other skills would? Establishing this would strengthen the case for generalizability and relevance.

Your question demonstrates remarkable expertise, and this submission process has been immensely beneficial, helping me better understand how to present empirical research logic. The selection of set shot as the target skill was primarily based on the following considerations:

I. Instructional Practice Relevance: Basketball is a compulsory component of the seventh-grade physical education curriculum in Chinese middle schools; it is widely popular among adolescent populations; and it facilitates the implementation of teaching experiments within regular physical education classes.

II. Motor Learning Characteristics: The focus is on gross limb segment fundamentals, offering advantages over fine motor skills (e.g., table tennis); technical evaluation is less constrained by observational limitations; and the movement staging is clearly defined.

III. Theoretical Compatibility: It aligns with the observational learning hypothesis of multimedia learning theory, as learners rely on visual demonstrations to establish operational representations; it possesses a distinct spatiotemporal movement structure; and it can be readily decomposed into discrete learning units.

Following your suggestion, we have added relevant descriptions in the main text to clarify: the logical connection between skill selection and the theoretical framework (CTML application scenarios); the scientific basis for movement staging; and the ecological validity of the instructional experiment design. Specific revisions are detailed below.

The set shot is a highly visual motor skill [44] that requires learners to establish internal representations by observing the movements, a process consistent with the fundamental assumptions of observational learning and imitation in multimedia learning theory. Additionally, this skill exhibits a well-defined spatiotemporal movement structure that can be clearly divided into preparation stage and execution stages and principally involves control of the upper-limb, the lower-limb, and the ball-handling. These characteristics render the set shot particularly well suited to examining the effects of movement staging and limb segment information guidance in phased motor skill instruction [44,45]. Finally, the set shot is a fundamental skill in secondary school physical education. Students typically lack experience with this skill, and its stable movement characteristics facilitate standardized assessment and instructional application.�Lines�167-177

1.3 Lastly, a central hypothesis or set of testable predictions is not explicitly stated. This makes it difficult to discern what the authors expected and how the results support or challenge their assumptions.

Thank you for pointing out the lack of clear hypotheses or testable predictions, which could indeed confuse readers and make it difficult to discern the paper's main focus at the outset. Following this suggestion, we have incorporated the following revisions in the Introduction section: (1) posing specific research questions to stimulate thinking on "which information presentation methods are truly effective for motor learning," and (2) stating testable predictions: "This study hypothesizes that...". Finally, the Conclusion section now explicitly states whether the results support the hypotheses. These suggestions have also enlightened us that adopting a reader-oriented perspective when writing academic papers can significantly improve the clarity of the research problems being addressed.

The recognition of these deficiencies in set shot pedagogy prompted the following research question: Which of the three information presentation methods—visual-only (video), dual-sensory audiovisual (video with narration), and guided dual-sensory instruction (video with narration and markers)—is most effective in motor learning?�Lines�181-184

This study hypothesized that incorporating verbal narration and visual markers (red flashing frames) in instructional videos would enhance learners’ focus on critical motor learning information, increasing movement pattern performance and motor cognition outcomes. This design is consistent with the core proposition of CLT, which posits that learners most effectively manage limited cognitive resources by focusing on the most task-relevant information.�Lines�189-194

The findings of this study validate the guided dual-sensory information strategy based on the dual-channel processing and signaling principles of the CTML.�Lines�767-768

2. Insufficient methodological details

2.1 The study does not provide adequate information about participant demographics beyond age (e.g., gender distribution, physical literacy or motor skill baseline beyond set shot experience). This is especially important given that sex or experience level could potentially influence motor learning or responsiveness to different instructional modalities.

Thank you for your suggestions on the details of this study. The basic information of the participants is very important, and we will carefully supplement the relevant content. The gender distribution has been supplemented, and the number of male and female students in the class is basically the same. In this experiment, under the premise of controlling the consistency of teaching process and content, the gender indicators of the subjects were not included in the measurement variables. 2. It should be acknowledged that physical literacy was not measured before the start of this experiment. This teaching class is divided into natural classes. Prior to the start of the experiment, we communicated with the physical education teacher and the homeroom teacher. Based on previous records, there was no significant difference in physical fitness among the three classes. According to the analysis of physical education grades, there is no difference in basic motor skills other than shooting among the three classes. These supplementary contents have been reflected in the research object section of the revised manuscript. We sincerely thank you for pointing out this key methodological issue, which significantly enhances the rigor of the research.

The participants were randomly assigned to one of three instructional conditions: a video only group (n = 44), a video with narration (audiovisual) group (n = 44), and a video with narration and markers (guided dual-sensory instruction) group (n = 44), with Classes A (video) and B (video with narration) each contributing 22 male and 22 female students, and Class C (video with narration and markers) contributing 21 male and 23 female students.�Lines�212-217

Before the experiment, all participants and their parents/guardians were fully informed of the study procedures and safety considerations, and written informed consent was obtained from both the participants and their parents/guardians. Three experts independently conducted baseline assessments of the participants’ motor skills. As detailed in the Results section, no significant between-group differences were observed in the pretest set shot performance among the three groups (p > 0.05), ensuring sample homogeneity.�Lines�224-229

During the experimental period, the participants did not engage in any additional basketball activities or any form of basketball skills training outside the study protocol.�Lines�234-236

2.2 Additionally, the randomization process is not described in sufficient detail. The authors state that participants were randomly assigned to one of three instructional groups, but there is no mention of how this was done (e.g., computer-generated random numbers, stratified sampling, class-based assignment). This limits confidence in the equivalence of groups at baseline.

We fully agree with the process description of supplementing random grouping. Thank you for your patient guidance. You have filled in the gaps in our research from the perspectives of experts and readers, and helped improve this article.

The study school admits students using a proximity-based enrollment policy and maintains regular mixed-gender classes. Under these conditions, the motor abilities of the students in the same grade approximate a normal distribution, satisfying random sampling criteria. We employed cluster sampling, randomly selecting three out of the nine seventh-grade classes. �Lines�208-212

During the experimental period, the participants did not engage in any additional basketball activities or any form of basketball skills training outside the study protocol.�Lines�234-236

2.3 It is also unclear whether the expert raters were blinded to group allocation during performance assessment. Without blinding, there is risk of observer bias, particularly since scoring was based on qualitative movement ratings.

You have conducted a very detailed review of this article, and expert raters have blindly tested the grouping information during the evaluation process. I have already supplemented that this part of the content is included in the learning effectiveness data collection section.

The final scores were determined by averaging the ratings of the three experts. The expert raters were blinded to the group assignment information during the evaluation process.�Lines�389-391

3. Ambiguity in describing the guided “marker” cues

3.1 The “guided dual sensory” condition (i.e. the addition of visual markers) is insufficiently described. The manuscript does not clarify what the markers looked like (e.g., flashing frames, arrows, highlights?), where on the screen they appeared (e.g., limb segments, ball), how long they were displayed, or whether they were dynamic (moving) or static. These features are critical to understanding how attention might have been guided.

This valuable feedback has significantly enhanced our presentation of relevant content. We acknowledge that the original description of markers lacked sufficient

---

## [Decision Letter · Decision Letter 1]

15 Oct 2025

Dear Dr. Chiu,

Thank you for submitting your manuscript to PLOS ONE. After careful consideration, we feel that it has merit but does not fully meet PLOS ONE’s publication criteria as it currently stands. Therefore, we invite you to submit a revised version of the manuscript that addresses the points raised during the review process.

I thank the authors for their attention to the reviewers' feedback. The manuscript has improved substantially due to the revisions implemented.  R1 raises some additional minor points that would further improve the manuscript. Thus, I am requesting that the authors submit an additional revision responsive to these points.  If the authors do so, I will review their responses and render a decision without sending the manuscript back out for an additional round of review.

We look forward to receiving your revised manuscript.

Kind regards,

Laura Morett

Academic Editor

PLOS ONE

Journal Requirements:

Reviewers' comments:

Reviewer's Responses to Questions

**Comments to the Author**

Reviewer #1: (No Response)

2. Is the manuscript technically sound, and do the data support the conclusions?

Reviewer #1: Partly

3. Has the statistical analysis been performed appropriately and rigorously?

Reviewer #1: Yes

4. Have the authors made all data underlying the findings in their manuscript fully available?

Reviewer #1: Yes

5. Is the manuscript presented in an intelligible fashion and written in standard English?

Reviewer #1: Yes

Reviewer #1: This revised version shows clear progress compared to the previous one. The presentation of the methodology is now much clearer, the statistical analyses are more rigorously applied, and the overall discussion appears more coherent and well-structured.

Abstract

Line 25 : It would seem more appropriate to place point (2) before point (1), moving from the more general to the more specific, and to avoid repeating the group comparison.

Method

Participants :

You do not specify how the randomization was carried out

Line 240 and 247 : You repeat « consent forms were signed by the participants’ parents »

Line 259 : Specify why the 0.8 threshold is chosen as a reference (often attributed to Cohen, 1988), as this would strengthen the justification.

Line 272 « Movement patterns and movement performance were assessed using a mixed-design ANOVA, … »

The 3 conditions are much better presented , which makes it easier for the reader to understand.

Line 320 : Could you justify the thresholds used (0.2; 0.4...)?

Figure

Could you make three squares of the same size for the three conditions and explain in the legend that each participant only completed one of the three conditions? For the time labels, sometimes the first letter is capitalized and sometimes not; please harmonize. It would be more appropriate if the retention test followed the post-test in a linear way (by enlarging the figure and continuing on the next page). The phrase “After 6 classes” is misplaced, as it gives the impression that six classes occur between the end of the protocol and the posttest session.

Learning outcome data collection

Line 389 and….: Could you provide examples of items or questions for each of the scales (motor cognition, movement result performance...)?

Results :

It would seem more appropriate to present the results in the form of graphs rather than tables.

Discussion :

Line 665-672 : Be careful in your interpretation, this remains a hypothesis.

Line 673-683 : The authors argue that visual cues enhance learning by guiding attention. Given that no direct measures of attention (e.g., eye-tracking, subjective scales) were used, this limits the robustness of the interpretations. I suggest qualifying the conclusions on this point and stating more explicitly that these results remain hypothetical.

Line 676 : « For example, Zhang et al. » the date is missing after the reference

Line 649 : You discuss multimodality as the contribution of complementary information, but in itself multimodality can improve motor performance and learning, as seen in these papers :

• Blais, M., Jucla, M., Maziero, S., Albaret, J. M., Chaix, Y., & Tallet, J. (2021). Specific cues can improve procedural learning and retention in developmental coordination disorder and/or developmental dyslexia. Frontiers in Human Neuroscience, 15, 744562.

• Lagarrigue, Y., Cappe, C., & Tallet, J. (2021). Regular rhythmic and audio-visual stimulations enhance procedural learning of a perceptual-motor sequence in healthy adults: A pilot study. PLoS One, 16(11), e0259081.

• Diederich A, Colonius H. Bimodal and trimodal multisensory enhancement: Effects of stimulus onset and intensity on reaction time. Perception & Psychophysics. 2004 Nov;66(8):1388–404. pmid:15813202

• Hecht D, Reiner M, Karni A. Multisensory enhancement: gains in choice and in simple response times. Exp Brain Res. 2008 May 14;189(2):133. pmid:18478210

**Do you want your identity to be public for this peer review?** For information about this choice, including consent withdrawal, please see our Privacy Policy

Reviewer #1: No

---

## [Author Response · Author response to Decision Letter 2]

27 Oct 2025

Dear Editor

Thank you for your insightful suggestions. Our team has carefully studied each of the revision suggestions for this article and made corresponding modifications. We hope that our responses address the reviewers' comments satisfactorily, and we would be pleased to consider any further suggestions. Thank you for your thorough review of this article once again.

Best regards,

The authors

The following is a revision explanation:

• Black text represents the original comments from the reviewers

• Purple text indicates our direct responses to the reviewers' comments

• Blue text shows the revised content that corresponds to changes made in the revised manuscript

Reviewer 1

Abstract

Line 25 : It would seem more appropriate to place point (2) before point (1), moving from the more general to the more specific, and to avoid repeating the group comparison.

Thank you for your suggestion and for reading this article so carefully. The correction has been completed.

(1) All three groups (visual-only, dual-sensory, and dual-sensory with visual markers) exhibited improved movement result performance after training, with the group receiving guided dual-sensory instruction exhibiting substantially superior posttest movement result performance than the group receiving video alone. (2) audiovisual (dual-sensory) information considerably enhanced movement pattern performance, and dual-sensory information with additional visual markers strengthened movement pattern retention and optimized upper-limb movement pattern performance during the preparation stage. �Lines�25-31

Method

Participants :

You do not specify how the randomization was carried out

The study school admits students using a proximity-based enrollment policy and maintains regular mixed-gender classes. Under these conditions, the motor abilities of the students in the same grade approximate a normal distribution, satisfying random sampling criteria.

We employed cluster sampling, randomly selecting three out of the nine seventh-grade classes. �Lines�227-228

Line 240 and 247 : You repeat « consent forms were signed by the participants’ parents »

The second occurrence of « consent forms were signed by the participants’ parents » has been deleted. Thank you for your careful review.

Line 259 : Specify why the 0.8 threshold is chosen as a reference (often attributed to Cohen, 1988), as this would strengthen the justification.

Relevant literature support has been added.

The achieved power of 0.87 met the 0.8 threshold [46]. �Lines�258

46. Cohen J. Set correlation and contingency tables. Applied psychological measurement. 1988; 12(4): 425-434.

Line 272 « Movement patterns and movement performance were assessed using a mixed-design ANOVA, … »

The 3 conditions are much better presented , which makes it easier for the reader to understand.

Thank you again for the meticulous correction. It has been completed.

Movement patterns and movement performance were assessed using a mixed design ANOVA, with……�Lines�271-272

Line 320 : Could you justify the thresholds used (0.2; 0.4...)?

Supplementary references have been provided to explain the applicable thresholds.

Items with D values < 0.2 should be eliminated, and items with D > 0.4 have high discriminatory ability and reliability and should be retained[49]. �Lines�391-320

Figure

Could you make three squares of the same size for the three conditions and explain in the legend that each participant only completed one of the three conditions? For the time labels, sometimes the first letter is capitalized and sometimes not; please harmonize. It would be more appropriate if the retention test followed the post-test in a linear way (by enlarging the figure and continuing on the next page). The phrase “After 6 classes” is misplaced, as it gives the impression that six classes occur between the end of the protocol and the posttest session.

Your suggestion is very good. We have made modifications to the figure.

Learning outcome data collection

Line 389 and….: Could you provide examples of items or questions for each of the scales (motor cognition, movement result performance...)?

Thank you for your revision suggestions. Due to limited space, we have chosen to provide examples to illustrate. In this experiment, the learning information provided in the video materials was consistent with both the Set Shot Movement Pattern Evaluation Scale and the Set Shot Motor Cognition Test, establishing alignment across learning materials, motor skill assessments, and motor cognition evaluations.

For example, the key point: In the preparation stage, the ball should be held between the chin and the chest. This is included as part of the multimedia teaching materials during instruction. It also appears as a multiple-choice question in the Set Shot Motor Cognition Test (e.g., "During the preparation stage, where should the ball be held?"). In the Movement Pattern Performance assessment, this is a scoring criterion. Expert raters observe the students' movements and evaluate them based on the key technical points. �Lines�393-398

Results :

It would seem more appropriate to present the results in the form of graphs rather than tables.

We thank the reviewer for the suggestion. We understand the advantages of figures in terms of visualization. Considering that the results of this study involve numerical comparisons across multiple dimensions (such as movement staging and movement limb segments), we believe that tables are currently the form that can most clearly and accurately present this structured data. We have already provided focused interpretation of the key findings from the tables in the main text to aid reader comprehension.

Discussion :

Line 665-672 : Be careful in your interpretation, this remains a hypothesis.

Line 673-683 : The authors argue that visual cues enhance learning by guiding attention. Given that no direct measures of attention (e.g., eye-tracking, subjective scales) were used, this limits the robustness of the interpretations. I suggest qualifying the conclusions on this point and stating more explicitly that these results remain hypothetical.

The above suggestions have been revised. Thank you again for your feedback.

This finding lends support to the hypothesis that…… �Lines�676

this study hypothesized that……�Lines�685

Line 676 : « For example, Zhang et al. » the date is missing after the reference

The correction has been completed.

Line 649 : You discuss multimodality as the contribution of complementary information, but in itself multimodality can improve motor performance and learning, as seen in these papers :

Thank you very much to the reviewer for providing many references. After reading, we have selected the following references for use.

Research has demonstrated that consistent audiovisual stimuli can facilitate procedural perceptual-motor learning[51]. �Lines�627-629

Research showed that multisensory stimulation not only facilitates early perceptual processing and motor responses but also enhances higher cognitive processes such as information discrimination and response selection [63].�Lines�757-759

51. Lagarrigue Y, Cappe C, Tallet J. Regular rhythmic and audio-visual stimulations enhance procedural learning of a perceptual-motor sequence in healthy adults: A pilot study. PLoS One. 2021;16(11): e0259081.

63. Hecht D, Reiner M, Karni A. Multisensory enhancement: gains in choice and in simple response times. Exp Brain Res. 2008 May 14;189(2):133.

---

## [Editor Report · Decision Letter 2]

5 Nov 2025

Effect of guided dual-sensory information on motor learning outcomes based on spatiotemporal dimensions

PONE-D-25-06505R2

Dear Dr. Chiu,

We’re pleased to inform you that your manuscript has been judged scientifically suitable for publication and will be formally accepted for publication once it meets all outstanding technical requirements.

Kind regards,

Laura Morett

Academic Editor

PLOS ONE

Additional Editor Comments (optional):

I thank the authors for revising the manuscript to address R1's remaining comments. I am now pleased to recommend the manuscript for publication in PLOS One.
---

## [Editor Report · Acceptance letter]

PONE-D-25-06505R2

PLOS ONE

Dear Dr. Chiu,

I'm pleased to inform you that your manuscript has been deemed suitable for publication in PLOS ONE. Congratulations! Your manuscript is now being handed over to our production team.

Kind regards,

on behalf of

Dr. Laura Morett

Academic Editor

PLOS ONE